# RNA Sequencing Reveals the Involvement of Serum Exosomal miRNAs in Early Pregnancy in Cattle

**DOI:** 10.3390/ani14172600

**Published:** 2024-09-06

**Authors:** Zhongxiang Ji, Binwu Bao, Yumei Wang, Zhengxing Wang, Yi Yang, Jinrui Xu, Xingping Wang, Zhuoma Luoreng

**Affiliations:** 1College of Animal Science and Technology, Ningxia University, Yinchuan 750021, China; 13083726975@163.com (Z.J.); b013019@126.com (B.B.); wym18238725837@126.com (Y.W.); wangzhengxing0707@163.com (Z.W.); 2School of Life Sciences, Ningxia University, Yinchuan 750021, China; yangyi@nxu.edu.cn (Y.Y.); xjr975@163.com (J.X.)

**Keywords:** bovine, pregnancy, exosomes, miRNA, embryo implantation, reproduction

## Abstract

**Simple Summary:**

Low fertility is the main cause of low productivity in beef cattle, and the non-conception of cows after fertilization is an important factor contributing to this phenomenon. miRNAs may exert unique metabolic and immune effects on the early gestation process in cattle. In this study, serum exosomes on the 0 th, 14 th, and 21 st days of the early gestation of cattle were obtained by simultaneous estrus technology, and the expression profile of exosomal miRNA was detected by RNA-seq technology. Among them, bta-miR-3604, bta-miR-2889, bta-miR-3432a, and bta-miR-409b are newly discovered miRNAs. This study can provide a theoretical reference for the screening of early pregnancy establishment and maternal pregnancy recognition (MRP) molecular markers in cattle.

**Abstract:**

Low fertility is the main cause of the low productivity in beef cattle and is mainly associated with a lack of conception after fertilization. The establishment of early pregnancy in cattle is a complex physiological process, and embryo implantation is crucial for the successful establishment of pregnancy. Exosomal miRNAs play an important role in regulating mammalian embryo implantation and development. This study used synchronous estrus technology to extract exosomes from bovine serum at 0, 14, and 21 days of early pregnancy and analyzed the expression profile of exosomal miRNAs through RNA-seq technology. We identified 472 miRNA precursor sequences and 367 mature miRNA sequences in the three sample groups, with the majority of the miRNAs having high abundance. Differentially expressed miRNAs (DEmiRNAs) were screened, and 20 DEmiRNAs were obtained. The differential expression analysis results show that compared to day 0, there were 15 DEmiRNAs in the serum on day 14 and 5 on day 21 of pregnancy. Compared to the 14th day of pregnancy, there were eight DEmiRNAs in the serum on the 21st day of pregnancy. Bioinformatics analysis shows that the target genes of DEmiRNAs regulated the signaling pathways closely related to early pregnancy, including the VEGF, NF-κB, and MAPK signaling pathways. In addition, the newly discovered miRNAs were bta-miR-3604, bta-miR-2889, bta-miR-3432a, and bta-miR-409b. These results provide a theoretical reference for screening the molecular markers for early pregnancy establishment and maternal recognition of pregnancy (MRP) in cattle and new ideas for shortening the calving interval in cows.

## 1. Introduction

Reproductive traits are one of the important economic traits in the cattle industry, and the level of cattle productivity directly affects the economic benefits of farms. In general, cows only have one calf at a time, and the pregnancy failure rate can be as high as 40–50% during the first month of pregnancy [1]. The low fertility rate is the main factor affecting the productivity of cattle and may occur due to several reasons, such as increased insemination times, the extension of empty pregnancy time, and an increased elimination rate after the failure of artificial insemination. These problems are not only related to the environment and feeding management of farms but also closely related to the gene regulation of individual cattle before and after pregnancy. Therefore, improving the reproductive efficiency of cattle is an important strategy to increase the economic benefits of farms [2].

The establishment of mammalian pregnancy is an evolutionarily conserved and complex physiological process. Successful pregnancy requires the normal development of the embryo itself and an intrauterine environment conducive to embryo implantation [3]. Embryo implantation is an important part of mammalian pregnancy, and the whole pregnancy process is accompanied by a series of dynamic changes, including cell–cell interactions and cell signal transduction. MiRNA is an endogenous non-coding RNA with a short chain length of about 17~25 nucleotides, which is highly conserved and stable in animal body fluid. MiRNA regulates gene expression by inhibiting the translation of messenger RNA by binding its 3′ untranslated region (3′UTR), which is highly complementary to the target gene [4]. This affects biological processes such as cell metabolism, proliferation, differentiation, apoptosis, and individual life development [5]. Studies have shown that miRNAs also regulate inflammatory responses, the development of endometrium before embryo implantation, and the expression of immune tolerance-related genes during the beginning and maintenance of pregnancy. Exosomes are small vesicles with an average diameter of about 40~100 nm and are secreted by eukaryotic cells into the extracellular environment via exocytosis. They have a lipid bilayer membrane structure and contain nucleic acids (such as miRNA), proteins, and bioactive substances, which can be used as a medium for intercellular communication and information transmission. Exosomes widely exist in animal body fluids, including blood, urine, follicular fluid, uterine fluid, and milk [6], and can transfer bioactive substances by fusing with host membrane proteins and target cell membrane proteins, thus playing a regulatory role on target cells. Therefore, exosomes are considered good biomarkers for human clinical diagnosis and have attracted the attention of researchers worldwide.

It has been found that exosomes play an important role in mammalian pregnancy. For example, uterine fluid exosomes can regulate the development and implantation of embryos [7]. Studies have shown that the mammalian placenta can help the maternal body adapt to the pregnancy process by secreting bioactive substances and exosomes as messengers between the mother and the fetus [8,9,10]. Blood miRNA also plays an important role in regulating pregnancy in dairy cows and has a potential role as a biomarker for early pregnancy diagnosis [11]. Some scholars have reported that exosomes are the main carriers of miRNA in bovine serum, accounting for 78% of the total miRNA expression in serum and thus could have specific biological functions. Blood exosomes are also involved in immune regulation, maternal pregnancy recognition, and embryo implantation during maternal pregnancy [12,13].

Plasma exosomal miRNAs have been shown to be candidate biomarkers for the diagnosis of early pregnancy in heifers [14]. However, little research has been undertaken on the role of serum exosomal miRNAs in early pregnancy in heifers. Therefore, this study used transcriptome sequencing (RNA-seq) technology to reveal the miRNA expression profile of exosomes at 0, 14, and 21 days of early pregnancy in cattle. The study also screened the differentially expressed miRNAs (DEmiRNAs) at different times of early pregnancy in cattle to explore their role in early pregnancy and provide a new reference for molecular breeding to shorten calving intervals in cattle.

## 2. Materials and Methods

### 2.1. Sample Collection

The animal samples used in this study were from Guyuan Fumin Agricultural Technology Co., Ltd. in Ningxia Hui Autonomous Region, China. At the beginning of the experiment, 41 Guyuan yellow cattle were diagnosed with pregnancy using ultrasound detection technology and rectal grasping technology. Finally, 9 healthy, disease-free, and non-pregnant Guoyuan yellow cattle cows were selected, and they had similar body shapes and identical feeding conditions. Nine cows were subjected to estrus synchronization [15]. The specific operation process is as follows: each cow was intramuscularly injected with 200 micrograms of Gonarellin (Ningbo Sansheng Biotechnology, Ningbo, China) on the first morning, and the estrus status of the cows was observed for the next few days. If the cows were not in estrus, 1 milligram of Chloroprost sodium (Ningbo Sansheng Biotechnology, Ningbo, China) was injected on the seventh morning. If the cows were still not in estrus at this time, another 200 micrograms of Gonarellin was injected on the ninth day. The estrus status of the cow was observed at any time, and when it was finally observed that the cow was in estrus, it was naturally mated with the same bull on the tenth day. The peripheral blood of all cows was collected through the jugular vein and centrifuged at 4000 rpm for 15 min at 0, 14, and 21 days after mating. The supernatant serum was collected and frozen. On the 28th day after mating, the pregnancy status of the cows was detected using early pregnancy test strips, and it was found that 6 out of 9 cows were pregnant, and 3 cows were not pregnant. Subsequently, serum samples collected from three pregnant cows on days 0, 14, and 21 of early pregnancy were randomly selected for extracellular vesicle isolation.

### 2.2. Extracellular Vesicle Isolation, RNA Library Construction, and RNA-Seq

Exosomes were isolated from all serum samples using the exorneasy Maxi Kit (Qiagen, Hilden, Germany) and used for total RNA extraction. The RNA purity (od260/280 ratio) was determined using a nanodrop (nanodrop technologies, Wilmington, NC, USA), and the RNA integrity was accurately detected using a highly sensitive Agilent 2100 (Agilent, Santa Clara, CA, USA). A small RNA (sRNA) library was constructed using the small RNA sample pre kit (small RNA sample pre kit, Lexogen, Vienna, Austria) according to the manufacturer’s instructions. After library construction, qubit 2.0 was used to preliminarily quantify the effective concentration of the library, and the library was diluted to 1 ng µL^−1^. The insert size of the library was detected using a highly sensitive Agilent 2100, and qPCR was used to accurately quantify the effective concentration of the library (>2 nm) to ensure the quality of the library. After the library was quantified according to the effective concentration and the requirements of data output, it was subjected to high-throughput sequencing on the Illumina se 50 platform (Beijing Nuohe Zhiyuan Technology Co., Ltd., Beijing, China).

### 2.3. miRNA Sequence Alignment and Identification

Bowtie software (http://bowtie.cbcb.umd.edu, accessed on 27 August 2024) [16] was used to align the screened sRNA on the reference sequence and determine the distribution of sRNA on the reference sequence. The bovine miRNAs found in the miRBase database were considered existing miRNAs, and the known miRNAs were identified by comparing them with those in the miRBase database. The miRNAs identified using the hairpin structure prediction and reference sequences were considered novel miRNAs.

### 2.4. Screening and Analysis of DEmiRNAs

The expression of miRNA was quantified via stringtie software (v1.3.3), and the expression levels of each transcript were calculated according to the frequency of clean reads. The expression levels of known and new miRNAs in each sample were counted, and the expression levels were normalized to transcript per million (TPM) values. The TPM was calculated as (readcount × 1,000,000)/libsize, where libsize is the sum of the sample miRNA read count. The DEmiRNAs were analyzed in the samples collected at 0, 14, and 21 days of early pregnancy using the cuff software (V2.1.1), with |log_2_ ^(Fold Change)^| > 1 and *p* < 0.05 as the standard.

### 2.5. Validation of RNA-Seq Data by qPCR

Seven DEmiRNAs identified via RNA-seq analysis were randomly selected. The stem ring method [17] was used to design specific reverse transcription primers for the DEmiRNAs, while primer 5.0 software was used to design fluorescent quantitative PCR (qPCR) primers for miRNAs (Table 1). The primer sequences were synthesized by General Biology (Anhui) Co., Ltd. in Chuzhou, China. The qPCR reaction mixture consisted of 10.0 µL of SYBR Premium EX Taq, 2.0 µL of cDNA, 0.8 µL of forward and reverse primers (10 µL mol L^−1^), and 6.4 µL of RNase free water. The qPCR reaction conditions were pre-denaturation at 95 °C for 5 min, followed by 35 cycles of denaturation at 95 °C for 30 s, annealing at 60 °C for 30 s, and extension at 70 °C for 60 s, and a final extension at 72 °C for 7 min and holding at 4 °C. Glyceraldehyde-3-phosphate dehydrogenase (GAPDH) and ribosomal protein S18 (RPS18) were used as internal reference genes, and the delta–delta Ct (2^−ΔΔCt^) method was used to calculate the relative expression levels of significantly different miRNAs [18]. Three technical replicates were set for each sample, and the results are presented as mean ± standard deviation (X ± SD)

### 2.6. Prediction of Target Genes and Functional Enrichment Analysis of DEmiRNAs

The target genes of DEmiRNAs were predicted using miranda-3.3a and rnahybridv2.0 software, and the corresponding relationships between the miRNAs and the target genes were determined. Clusterprofiler software was used for the gene ontology (GO) function enrichment of the DEmiRNA target genes, including the molecular function (MF), biological process (BP), and cellular component (CC). Clusterprofiler (3.8.1) software was used to analyze the signal pathway of the DEmiRNA target genes using the Kyoto Encyclopedia of Genes and Genomes (KEGG) database. The biological functions of DEmiRNAs were then obtained based on the results of GO and KEGG enrichment analyses.

### 2.7. Statistical Analysis

Three biological replicates and three technical replicates were set for each sample, and all data are expressed as the ‘mean ± standard deviation’ (x ± SD). The t-test analysis of GraphPad 8.3 software was used to analyze the significance of differences in data between groups. * (*p* < 0.05) indicates significant differences, and ** (*p* < 0.01) indicates highly significant differences.

## 3. Results

### 3.1. Quality Analysis of the Sequencing Data

Qualified sRNA libraries were reverse-transcribed into cDNA, and nine cDNA libraries were sequenced. Samples of RNA samples were isolated from three pregnant cows on days 0, 14, and 21 of pregnancy. The raw sequencing data were analyzed for quality control and filtered after which high-quality clean reads were obtained. The results show that the filtered data had an error rate of 0.01%, Q20 of >99.42%, Q30 of >96.99%, and GC content of 51.60–52.73%, indicating the high quality of sequencing data (Table 2). Moreover, the sequence alignment results show that more than 75.87% of the data in the nine groups of samples aligned to the reference sequence (Table 3). These findings show that the sequencing results were reliable and accurate and met the requirements for subsequent analysis.

### 3.2. Identification and Overall Distribution of miRNAs

After comparison with the miRbase database and reference sequences, 367 miRNA mature sequences and 472 miRNA precursor sequences were identified in nine samples (Table 4), of which 451 miRNA precursor sequences corresponded to mature sequences. In addition, 10 new miRNA precursor sequences were predicted, and 20 new miRNA precursor sequences corresponded to mature sequences. Nine new miRNA mature sequences were compared. To ensure the reliability and rationality of the experiment, we analyzed the correlation of miRNA expression levels between the samples (Figure 1). We found a correlation coefficient of around 0.6, which indicates a significant correlation between all samples. A density distribution map was constructed for each sample based on TPM (Figure 2), and the distribution and expression symmetry of all miRNAs were evaluated. The results show that the expression distribution of miRNAs in each sample was correct. Moreover, the distribution of miRNA expression levels between the samples was similar and suitable for further analysis.

### 3.3. Selection of DEmiRNAs

Through differential expression analysis, 20 unrepeated DEmiRNAs (Table 5) were identified in the three sample groups. Compared with day 0, there were 15 DEmiRNAs in serum exosomes on day 14 (a) after pregnancy, of which 12 were upregulated and 3 were downregulated, while 5 DEmiRNAs were identified in serum exosomes on day 21 (b) after pregnancy, of which 3 were upregulated and 2 were downregulated. Compared with the 14th day of pregnancy, there were eight DEmiRNAs in the exosomes on the 21st day of pregnancy (c), of which two were upregulated and six were downregulated. As shown in the Venn diagram (d), eight specific DEmiRNAs were observed between 0 days and 14 days of pregnancy, which are bta-miR-122, bta-miR-133a, bta-miR-185, bta-miR-202, bta-miR-2889, bta-miR-29a, bta-miR-3432a, and bta-miR-493, and two specific DEmiRNAs, bta-miR-128 and bta-miR-486, were observed on day 0 versus day 21 of pregnancy, and two specific DEmiRNAs, bta-miR-145 and bta-miR-409b, were observed on day 0 versus day 21 of pregnancy (Figure 3). Further analysis shows that bta-miR-1298, bta-miR-199a-3p, bta-miR-3604, bta-miR-218, bta-miR-1, bta-miR-199a-5p, bta-miR-107, and bta-miR-205 were expressed in the exosomes at 0, 14 and 21 days of pregnancy. Compared with the first day of pregnancy, the expression of bta-miR-199a-5p was continuously downregulated on the 14th and 21st day of pregnancy, while the expression of bta-miR-107 was upregulated on the 14th and 21st day of pregnancy, and the expression was the highest on the 14th day of pregnancy.

### 3.4. qPCR Validation of the RNA-Seq Data

To verify the accuracy and reliability of the RNA-seq data, we selected seven DEmiRNAs (three upregulated and four downregulated) for qPCR analysis. The qPCR detection results of the seven DEmiRNAs were consistent with the differential expression trend of the RNA-seq analysis results in each group (Figure 4), indicating that the results of RNA-seq analysis and differential expression analysis of serum exosome miRNAs were reliable.

### 3.5. Functional Enrichment Analysis of Target Genes of DEmiRNAs

Miranda (https://bibiserv.cebitec.uni-bielefeld.de/rnahybrid/submission.html/, accessed on 27 August 2024) and rnahybrid (http://mirtoolsgallery.tech/mirtoolsgallery/node/1055, accessed on 27 August 2024) software were used to predict the potential target genes of miRNA among the three groups. The prediction results show 893, 249, and 177 unreplicated target genes for 15 DEmiRNAs at 0 vs. 14 days of pregnancy, 5 DEmiRNAs at 0 vs. 21 days of pregnancy, and 8 DEmiRNAs at 14 vs. 21 days of pregnancy, respectively. The GO enrichment analysis shows that at the BP level, these target genes were significantly enriched in terms of signal regulation, positive regulation of gene expression, receptor binding regulation, and positive regulation of cell metabolic process. At the CC level, the target genes were significantly enriched in the cell cortex, Golgi apparatus, plasma membrane, and nucleus. However, at the MF level, these target genes were significantly enriched in the binding of transcriptional core compression, binding of regulatory region nucleic acid, protein binding, GTPase binding, cation transport ATPase activity, and other terms (Figure 5, Appendix A, Appendix A). These GO terms are related to embryonic development and implantation. The KEGG enrichment results show that the target genes were mainly significantly enriched in VEGF, NF-κB, MAPK, and other signaling pathways (Figure 6, Appendix A, Appendix A).

## 4. Discussion

Early pregnancy is a critical period for the establishment of mammalian pregnancy, and embryo implantation is crucial for successful pregnancy. This process requires blastocysts with implantation ability and good endometrial receptivity [19]. Exosomes are tiny vesicles secreted by various types of cells under normal or abnormal physiological conditions and are widely found in bodily fluids (such as saliva, breast milk, blood, follicular fluid, and uterine fluid) that contain various biological components, such as proteins, lipids, coding genes, and non-coding genes. Exosomes can serve as a medium for exchanging materials and information between cells [20]. Previous studies have shown that the uterine cavity, blood, follicular cavity, and placental exosomes of humans, mice, pigs, cows, and sheep may regulate several developmental processes, such as fertilization, embryonic development, embryo transfer, embryo implantation, and endometrial receptivity, and affect the immune processes related to endometrial implantation [21,22]. MiRNA is an important component of exosomes, which can control many important physiological and pathological processes and plays a crucial regulatory role in cell development and differentiation [23,24]. Studies have shown that miRNAs exhibit significantly different expression levels among different species and physiological stages, especially during pregnancy in mammals, where they regulate embryonic development, implantation, and tissue and organ formation [25]. Klohonatz, K.M. et al. [26] isolated exosomes from the serum samples of pregnant and non-pregnant mares collected at 12, 14, 16, and 18 days and conducted endometrial biopsy to analyze the expression of miRNAs. The results show that 12 DEmiRNAs were identified at specific stages, and maternal recognition of pregnancy (MRP) was used as a biomarker. Ioannidis et al. [27] found that miR-26a was significantly upregulated in pregnant cows on the 16th day of pregnancy compared with non-pregnant cows and considered it a potential biomarker for early pregnancy. However, among the DEmiRNAs identified in this study, only a few miRNAs were the same, except miR-29c, miR-101, and miR-30c. This indicates that the DEmiRNAs related to early pregnancy are not consistent among species and may be related to species, varieties, or individual physiological states. Among the 20 DEmiRNAs identified in this study, bta-miR-3604, bta-miR-2889, bta-miR-3432a, and bta-miR-409b were newly discovered miRNAs that may be related to early pregnancy. MiR-185, miR-1, and miR-107 have been reported to regulate early pregnancy [28]; for example, miR-185 plays an important regulatory role in early pregnancy. Studies have shown that miR-185 can affect the release of fetal placenta by fusing the VEGFA receptor with downstream activated PLC through the vascular endothelial growth factor A (VEGFA) signal pathway [29]. As the target gene of miR-185, matrix-interacting molecule 1 (STIM1) can affect the release of the placenta by regulating the concentration of Ca2+ in uterine cancer epithelial (UCE) cells [30].

Studies have shown that the VEGF, NF-κB, and MAPK signaling pathways play important roles in mammalian embryo implantation and embryonic development. In addition, miRNAs often regulate biological processes by modulating the target genes and the signaling pathways involved. To further explore the role of DEmiRNAs, we predicted their target genes and their functions. We found that 20 DEmiRNAs were mainly concentrated in the VEGF, NF-κB, MAPK, and other signaling pathways.

Angiogenesis is an important component of the embryo implantation process, and it is mainly mediated by VEGF [31]. Studies have shown that sorbin plays an important role in mediating angiogenesis for the successful pregnancy of mice by activating the VEGF signaling pathway and inhibiting the decline of endometrial receptivity in mice after sorbin administration causes embryo implantation failure [32]. For human beings, the VEGF signaling pathway plays a key role in the induction and differentiation of the mesoderm of embryonic stem cells (hESCs) and regulates the formation of various primitive embryogenic organs [33]. For cattle, VEGF can promote early embryonic development and oocyte maturation in vitro. VEGF is also involved in creating the best local environment for fertilization and bovine embryo development during the development of follicles by regulating permeability in the fallopian tube [34]. It has been reported that vegfr-3-deficient embryos show vascular system abnormalities and severe anemia. Furthermore, the VEGF-C signal transmitted through VEGFR-3 plays an important role in angiogenesis [35], which is conducive to the smooth implantation of embryos. Toll-like receptor 4 (TLR4), a membrane receptor and an activator of NF-κB, is widely expressed at the maternal–fetal interface. It produces inflammatory factors, such as tumor necrosis factor A (TNFa), interleukin 6 (IL-6), and interleukin 1B (IL1B) through NF-κB and affects the physiological environment of embryo implantation and uterus [36]. It has been reported that adding progesterone (P4) to bovine endometrial epithelial cells downregulates the expression of TLR4, NF-κB, and IL-6, thereby inhibiting the NF-κB signaling pathway. This indicates that P4 can regulate cell inflammation through the NF-κB and LIF/STAT3 pathways, reduce the contractile activity of the uterus during pregnancy, inhibit the maternal immune rejection of the fetus, prevent estrus, and promote uterine vasodilation and embryo implantation [37]. HMGB1 is related to placental validation and affects placental development throughout pregnancy. It can stimulate bovine ovarian granulosa cells to secrete IL-6 by activating NF-κB through TLR2 and regulate the expression levels of VEGF, EGFR, and other genes to further induce follicular development and ovulation, thus affecting embryo formation [38]. Studies have shown that interferon (IFN) is the initial pregnancy signal of ruminant embryos and can induce immune tolerance in humans and other species. By weakening the hmgb1/NF-κB signaling pathway, IFN can resist endometritis induced by *Escherichia coli* endotoxin, enhance endometrial receptivity, and promote embryo implantation [39]. The MAPK signaling pathway also plays an important role in regulating pregnancy maintenance, placental development, and fetal growth [40]. For pigs, the MAPK signaling pathway is involved in the signal transduction induced by VEGF during peri-implantation. During early pregnancy, VEGF affects the embryonic development of pigs by simultaneously stimulating the pi3k-akt1 and MAPK signaling pathways [41]. OPRM1 exists in mouse oocytes and granulosa cells, and it mediates the transitions of oocytes into the blastocyst stage and promotes embryo formation by regulating the MAPK pathway in oocytes [42]. For cattle, IGF1 mediated by the MAPK signaling pathway promotes embryo development into the blastocyst stage, which is crucial for the development of embryos before implantation [43]. EGF stimulates the proliferation of bovine cotyledon trophoblast cells in the in vitro model of the bovine placenta by activating the Ras and MAPK signal transduction pathways, thereby affecting the interaction between the mother and fetus during pregnancy [44].

Notably, bta-miR-3604, bta-miR-2889, bta-miR-3432a, and bta-miR-409b may act on signaling pathways, such as VEGF, NF-κB, and MAPK, through their target genes, thereby affecting early pregnancy-related processes, including embryo implantation and development. However, their specific molecular functions and regulatory mechanisms need to be further explored.

## 5. Conclusions

A total of 20 DEmiRNAs were obtained in serum exosomes of cattle on the 0th, 14th, and 21st day of pregnancy. The target genes were significantly enriched in signaling pathways, such as VEGF, NF-κB, and MAPK. In addition, these signaling pathways play an important role in gestation in yellow cattle. bta-miR-3604, bta-miR-2889, bta-miR-3432a, and bta-miR-409b are not only newly identified miRNAs but also have a high differential expression status on gestation day 14 or 21 compared to gestation day 0, suggesting that they can be used as an indicator for determining the early pregnancy rate in bovines. However, their functions and specific molecular regulatory mechanisms need further exploration.

## Figures and Tables

**Figure 1 animals-14-02600-f001:**
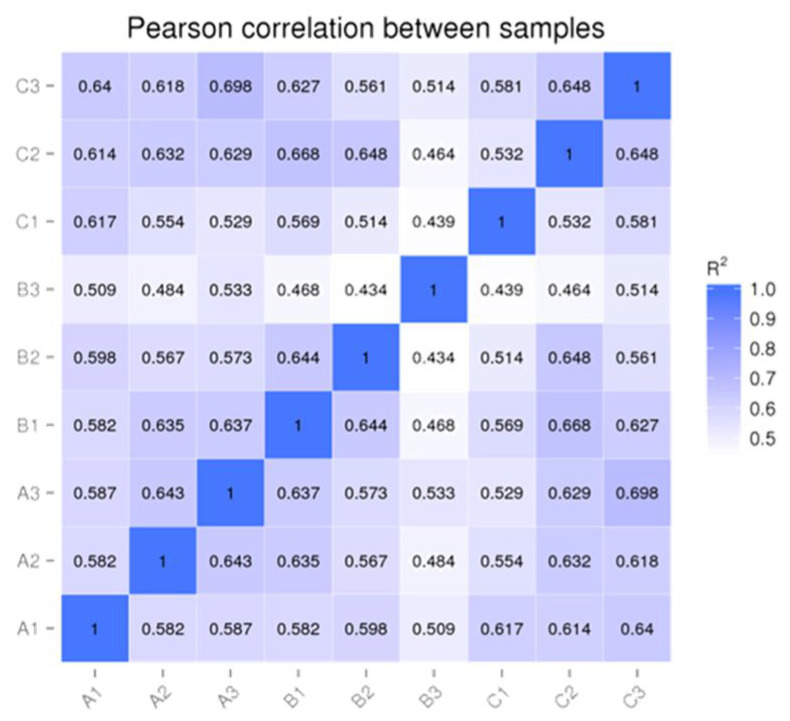
Correlation plots of miRNAs between samples.

**Figure 2 animals-14-02600-f002:**
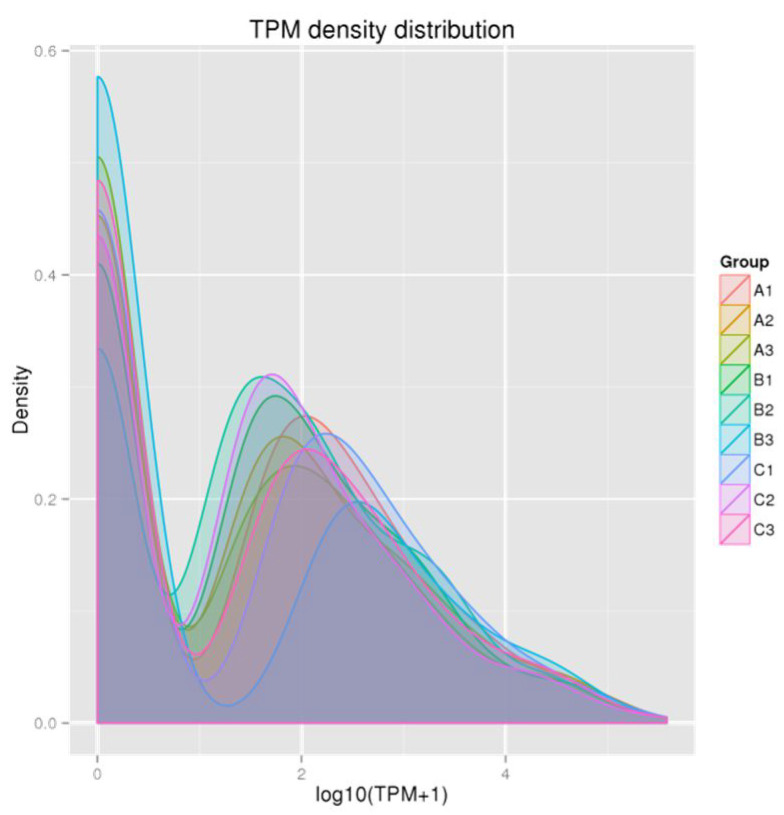
Density distribution pattern of transcript per million (TPM) values of all samples.

**Figure 3 animals-14-02600-f003:**
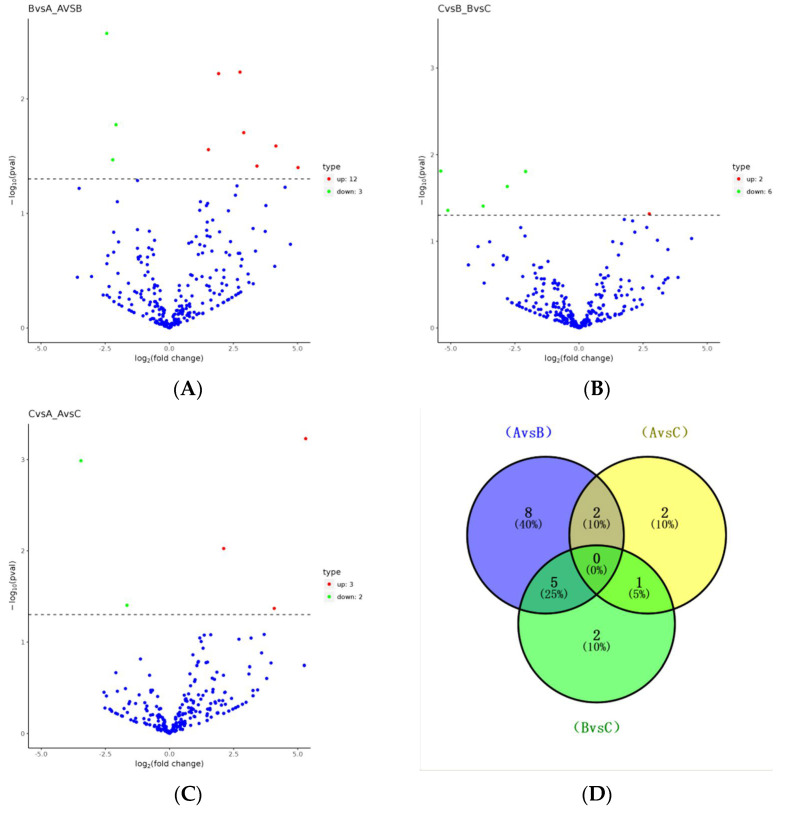
Volcano and Venn diagrams of the differentially expressed miRNAs (DEmiRNAs). (**A**) 14 d vs. 0 d; (**B**) 21 d vs. 0 d; (**C**) 21 d vs. 14 d; (**D**) Venn diagram.

**Figure 4 animals-14-02600-f004:**
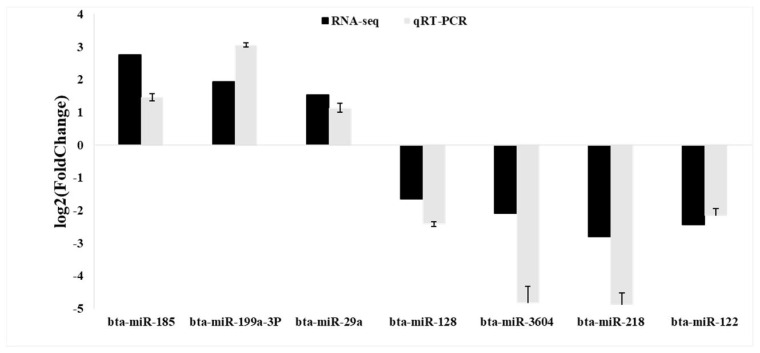
The qRT-PCR validation of the differentially expressed miRNAs (DEmiRNAs).

**Figure 5 animals-14-02600-f005:**
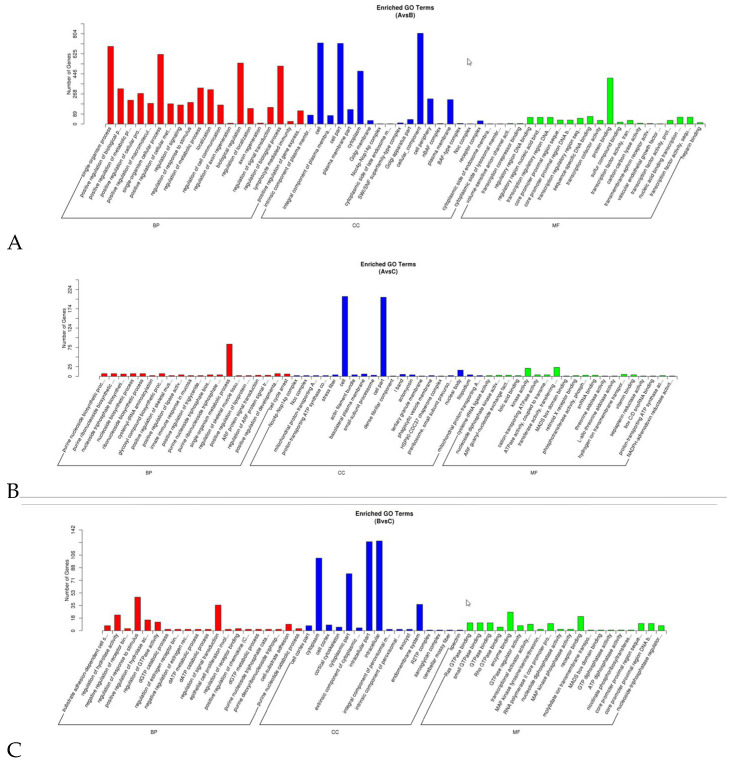
Gene ontology (GO) functional annotation of the target genes of differentially expressed miRNAs (DEmiRNAs). (**A**) 14 d vs. 0 d; (**B**) 21 d vs. 0 d; (**C**) 21 d vs. 14 d.

**Figure 6 animals-14-02600-f006:**
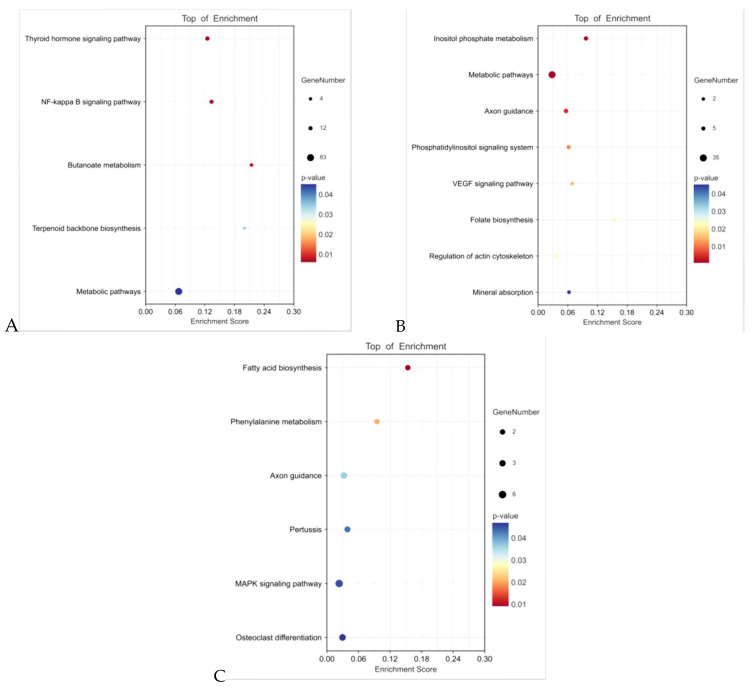
Kyoto Encyclopedia of Genes and Genomes (KEGG) functional analysis of the target genes of differentially expressed miRNAs (DEmiRNAs). (**A**) 14 d vs. 0 d; (**B**) 21 d vs. 0 d; (**C**) 21 d vs. 14 d.

**Table 1 animals-14-02600-t001:** Sequences of the reverse transcription and qPCR primers of DEmiRNAs.

miRNA Name	Reverse Transcription Primer Sequence (5′→3′)	qPCR Primer Sequence (5′→3′)	Product Length (bp)
bta-miR-185	RT:GTCGTATCCAGTGCGTGTCGTGGAGTCGGCAATTGCACTGGATACGACTCAGGAAC	F:GAATACTGGAGAGAAAGGCA R:GCAATTGCACTGGATACG	48
bta-miR-199a-3p	RT:GTCGTATCCAGTGCGTGTCGTGGAGTCGGCAATTGCACTGGATACGACAACAGGTA	F:GCTCGCCCCAGTGTTCAGAC R:CGTGTCGTGGAGTCGGCA	62
bta-miR-29a	RT:GTCGTATCCAGTGCGTGTCGTGGAGTCGGCAATTGCACTGGATACGACTAACCGAT	F:GCCTCACTAGCACCATCTGAA R: TATCCAGTGCGTGTCGTG	73
bta-miR-128	RT:GTCGTATCCAGTGCAGGGTCCGAGGTATTCGCACTGGATACGACAAAGAGAC	F:TGCTCATCACAGTGAACCG R:AGTGCAGGGTCCGAGGTATT	62
bta-miR-3604	RT:GTCGTATCCAGTGCAGGGTCCGAGGTATTCGCACTGGATACGACACAGTAGT	R:GCCTCATAACCAATGTGCAG F:AGTGCAGGGTCCGAGGTATT	63
bta-miR-218	RT:GTCGTATCCAGTGCAGGGTCCGAGGTATTCGCACTGGATACGACCACATGGT	F:GCAGCATTGTGCTTGATCTA R: GAGGTATTCGCACTGGATA	51
bta-miR-122	RT:GTCGTATCCAGTGCGTGTCGTGGAGTCGGCAATTGCACTGGATACGACCAAACACC	F:GCAGCATGGAGTGTGACAAT R:TATCCAGTGCGTGTCGTG	72

**Table 2 animals-14-02600-t002:** Quality control of the raw sequencing data.

Group	Sample	Reads	Bases	Error Rate	Q20	Q30	GC Content
0 d	A1	10,824,395	0.541 G	0.01%	99.46%	97.45%	52.56%
	A2	11,291,469	0.565 G	0.01%	99.52%	98.04%	51.60%
	A3	10,828,966	0.541 G	0.01%	99.57%	98.20%	52.43%
14 d	B1	11,827,970	0.591 G	0.01%	99.48%	97.72%	52.63%
	B2	11,464,028	0.573 G	0.01%	99.54%	97.94%	51.85%
	B3	15,418,305	0.771 G	0.01%	99.50%	97.80%	52.57%
21 d	C1	10,487,519	0.524 G	0.01%	99.42%	96.99%	52.73%
	C2	10,822,832	0.541 G	0.01%	99.58%	98.58%	52.35%
	C3	12,057,138	0.603 G	0.01%	99.53%	97.99%	52.62%

Note: A1, A2, and A3 represent the three samples collected on the 0th day of pregnancy; B1, B2, and B3 represent the three samples collected on the 14th day of pregnancy; C1, C2, and C3 represent the three samples collected on the 21st day of pregnancy.

**Table 3 animals-14-02600-t003:** Comparison of the clean reads with bovine reference genomes.

Group	Sample	Total sRNA	Mapped sRNA	“+”Mapped sRNA	“-”Mapped sRNA
	A1	10,211,425 (100%)	8,058,781 (78.92%)	5,475,561 (53.62%)	2,583,220 (25.30%)
0 d	A2	10,898,020 (100%)	9,768,913 (89.64%)	6,700,448 (61.48%)	3,068,465 (28.16%)
	A3	9,859,695 (100%)	7,924,069 (80.37%)	5,629,459 (57.10%)	2,294,610 (23.27%)
	B1	11,478,340 (100%)	8,827,964 (76.91%)	5,880,868 (51.23%)	2,947,096 (25.68%)
14 d	B2	11,151,053 (100%)	8,976,220 (80.50%)	5,902,948 (52.94%)	3,073,272 (27.56%)
	B3	15,246,813 (100%)	1,300,366 (85.29%)	9,797,584 (64.26%)	3,206,078 (21.03%)
	C1	10,174,282 (100%)	7,719,617 (75.87%)	5,055,172 (49.69%)	2,664,445 (26.19%)
21 d	C2	10,430,486 (100%)	8,148,185 (78.12%)	5,504,272 (52.77%)	2,643,913 (25.35%)
	C3	10,137,855 (100%)	7,906,125 (77.99%)	5,336,579 (52.64%)	2,569,546 (25.35%)

**Table 4 animals-14-02600-t004:** Summary of miRNA information.

Group	Sample	Known miRNA Matrices	Known miRNA Precursors	Novel miRNA Matrices	Novel miRNA Precursors
0 d	A1	210	277	5	5
	A2	207	255	3	3
	A3	192	239	2	2
14 d	B1	228	292	3	3
	B2	257	342	4	4
	B3	143	190	2	2
21 d	C1	195	251	5	5
	C2	224	289	3	3
	C3	193	257	3	4
	Total	367	472	9	10

**Table 5 animals-14-02600-t005:** Screening for differentially expressed miRNAs.

Group	miRNA Name	log_2_ Fold Change	pval
0 d vs. 14 d			
	bta-miR-1	−3.419840451	0.0386019
	bta-miR-107	−4.157990081	0.025785178
	bta-miR-122	2.435598687	0.002682208
	bta-miR-1298	−5.783528305	0.010463213
	bta-miR-133a	−6.184832579	0.006337687
	bta-miR-185	−2.760173877	0.005839627
	bta-miR-199a-3p	−1.924179333	0.006023499
	bta-miR-199a-5p	2.203766067	0.034036793
	bta-miR-202	−5.021631332	0.039784364
	bta-miR-218	−2.904624102	0.019721384
	bta-miR-2889	−6.002972001	0.011132547
	bta-miR-29a	−1.52695064	0.02776566
	bta-miR-3432a	2.075160579	0.016811504
	bta-miR-3604	−1.924179333	0.006023499
	bta-miR-493	−6.028330221	0.006258652
0 d vs. 14 d			
	bta-miR-107	−4.08341007	0.042774401
	bta-miR-128	1.658418829	0.039524034
	bta-miR-199a-5p	3.454122921	0.001027989
	bta-miR-205	−5.30962881	0.000588845
	bta-miR-486	−2.112590613	0.009433274
14 d vs. 21 d			
	bta-miR-1	3.738895476	0.039250249
	bta-miR-1298	5.390119257	0.015470609
	bta-miR-145	5.113806788	0.043988941
	bta-miR-199a-3p	2.08290067	0.015652709
	bta-miR-205	−6.652315352	0.000397973
	bta-miR-218	2.796429934	0.023303866
	bta-miR-3604	2.08290067	0.015652709
	bta-miR-409b	−2.740989268	0.048214807

## Data Availability

Data is contained within the article or Appendix A.

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
