# Peer review of "RNA Sequencing Reveals the Involvement of Serum Exosomal miRNAs in Early Pregnancy in Cattle"

_animals, 2024, doi:10.3390/ani14172600_

Round 1

Reviewer 1 Report

Comments and Suggestions for Authors

General comments:

The study aimed to reveal the miRNA expression profile of exosomes at 0, 14, and 21 days of early pregnancy in cattle and screened the differentially expressed miRNAs (DEmiRNAs) at different times of early pregnancy in cattle to explore their role in early pregnancy

New and current information is presented. However, the manuscript would be improved if a specific hypothesis is established. Tables and figures should be prepared to be understood without reference to information in the body of the manuscript. The Conclusions must be improved following the hypothesis and objective stated.

Comments on the Quality of English Language

None

Author Response

Response to Reviewer 1 Comments

The study aimed to reveal the miRNA expression profile of exosomes at 0, 14, and 21 days of early pregnancy in cattle and screened the differentially expressed miRNAs (DEmiRNAs) at different times of early pregnancy in cattle to explore their role in early pregnancy.

Point 1:New and current information is presented. However, the manuscript would be improved if a specific hypothesis is established. Tables and figures should be prepared to be understood without reference to information in the body of the manuscript. The Conclusions must be improved following the hypothesis and objective stated.

Response 1: Thanks for your comments .We have added new graphical illustrations to the results section and marked some errors in red by detecting them ourselves.

Reviewer 2 Report

Comments and Suggestions for Authors

Comments and Suggestions for Authors

After reviewing the manuscript entitled “RNA-seq reveals the involvement of serum exosomal miRNAs in early pregnancy in cattle”, the following suggestions were made it. The manuscript is well-written and organized in all its sections. Only a few minor changes must be made before the manuscript is accepted for publication. Below are my specific comments:

Abstract

Lines 27-32: The information shown in these lines is not adequate. Additionally, all significant findings should be aggregated, and an appropriate significance value should be included whenever an effect or absence of effects is indicated.

Keywords: pregnancy, exosomes, miRNA, RNA-seq. These words used as keywords are the same as those previously used in the title of the manuscript. Keywords should be different from those in the title (but related to the topic) to broaden the reach of academic search engines in case the manuscript is later published.

Introduction

Lines 40-74: The introduction is well organized and written and contains recent and relevant scientific references.

Line 76: Based on the reviewed background, the authors should add a clear hypothesis before mentioning the objective of the study.

Material and methods

Lines 82-141: The materials and methods section is well-written and organized. All procedures are replicable and supported by scientific references.

Line 143: The authors must indicate which statistical tests they used to evaluate the data's normality and homogeneity of variance. This is very important to know if the correct tests were used.

Results

Lines 146-224: The description of the results is not clear and should be completely rewritten. Which treatment increased, decreased, or was not affected should be clearly stated and this description should be supported by the corresponding p-value. 

On the other hand, it is worth mentioning that the sequence, organization and graphics used to show the results obtained are adequate.

Discussion

Lines 225-294: The discussion section is well organized and contains adequate depth and scientific references. I have no suggestions in this section.

Conclusion

Lines 296-301: The conclusions are supported by the findings obtained. I have no suggestions in this section.

Comments on the Quality of English Language

The quality of the English is good, only minor to moderate changes required.

Author Response

Response to Reviewer 2 Comments

Point 1: Lines 27-32:The information shown in these lines is not adequate. Additionally, all significant findings should be aggregated, and an appropriate significance value should be included whenever an effect or absence of effects is indicated.

Response 1: Thanks for your comments . We've rounded up the information to make it complete.

Point 2: Keywords:pregnancy, exosomes, miRNA, RNA-seq.These words used as keywords are the same as those previously used in the title of the manuscript. Keywords should be different from those in the title (but related to the topic) to broaden the reach of academic search engines in case the manuscript is later published.

Response 2: Thanks for your comments.We have made the keywords as required.

Point 3: Line 76:Based on the reviewed background, the authors should add a clear hypothesis before mentioning the objective of the study.

Response 3: Thanks for your comments.We have provided a hypothesis in advance of the study objectives.

Point 4: Line 143:The authors must indicate which statistical tests they used to evaluate the data's normality and homogeneity of variance. This is very important to know if the correct tests were used.

Response 4: Thanks for your comments.We have modified the statistical analysis section and highlighted it in red.

Point 5: Lines 146-224The description of the results is not clear and should be completely rewritten. Which treatment increased, decreased, or was not affected should be clearly stated and this description should be supp:orted by the corresponding p-value.

On the other hand, it is worth mentioning that the sequence, organization and graphics used to show the results obtained are adequate.

Response 5: Thanks for your comments.We've rewritten the results, added graphs and tables and labelled them in red

Reviewer 3 Report

Comments and Suggestions for Authors

General description

The manuscript under review uses RNAseq to investigate the differential expression of exosomal (extracellular vesicles) miRNA in serum samples from three pregnant cows. The samples were taken on days 0, 14, and 21 from mating. The three pregnant cows were randomly sampled from a total of nine.

The known bta-miRNAs were found using the miRbase, and structural characteristics were used to find new miRNAs.

The authors describe differentially expressed miRNAs and use the target gene by such miRNAs to discuss the potential impacts on pregnancy physiology.

Critical review

1 – The provider of the commercial kit used to extract miRNA makes clear that it does not purify only exosomes. The better way and in tune with International Society of Extracellular Vesicles guidelines might be using “extracellular vesicles miRNAs” instead of exosomes, a subtype of extracellular vesicles.

2 – in lanes 167,168 and 169 the authors stated:

“To ensure the reliability and rationality of the experiment, we analyzed the correlation of 167 miRNA expression levels between the samples (Fig. 1). We found a correlation coefficient of around 0.7, indicating a 168 relatively close relationship between the samples in each group”

However, if we examine Figure One in-depth, the numbers do not support the above statement. There is no pairwise comparison that shows R greater than 0.7. The numbers give a correlation coefficient between 0.5 and 0.6.

3—The authors use inadequate reference genes for the qPCR of miRNAs. Neither GAPDH nor RPS18 is recommended for such applications. Small Non-Coding RNAs or Small Nucleolar RNAs (snoRNAs) are preferred. Using spikes in miRNA to control miRNA extraction efficiency is also important. Its important to the authors to check the Minimum Information for Publication of Quantitative Real-Time PCR Experiments (MIQE) guidelines (https://academic.oup.com/clinchem/article/55/4/611/5631762)

4—The authors' path chosen to lead the discussion might be reevaluated. Most of the discussion involves physiological processes based on genes that interact with miRNA; however, the manuscript does not deal with functional experiments. Most of the discussion deals with speculative impacts that are not supported by experiments developed in the context of the manuscript. The authors do not clearly describe the putative consequential role played by extracellular vesicle miRNAs if they showed to be up or downregulated.

5 – in lane 184: Venn instead of “Wayne diagram”

Conclusion

The manuscript brings interesting data to the field of animal reproduction, which would be more relevant if the authors included data from non-pregnant animals in the analysis. Doing so would make it possible to get a glimpse of potential biomarkers related to pregnancy success. In the presented form, it is hard to figure out the relevance of blood serum extracellular vesicle miRNAs differentially expressed.

Author Response

Response to Reviewer 1 Comments

Point 1: The provider of the commercial kit used to extract miRNA makes clear that it does not purify only exosomes. The better way and in tune with International Society of Extracellular Vesicles guidelines might be using “extracellular vesicles miRNAs” instead of exosomes, a subtype of extracellular vesicles.

Response 1: Thanks for your comments.We did this by first extracting exosomes from serum samples via an exosome kit, followed by subsequent experiments with the returned RNA samples.

Point 2: in lanes 167,168 and 169 the authors stated:“To ensure the reliability and rationality of the experiment, we analyzed the correlation of 167 miRNA expression levels between the samples (Fig. 1). We found a correlation coefficient of around 0.7, indicating a 168 relatively close relationship between the samples in each group”However, if we examine Figure One in-depth, the numbers do not support the above statement. There is no pairwise comparison that shows R greater than 0.7. The numbers give a correlation coefficient between 0.5 and 0.6.

Response 2: Thanks for your comments.I've changed the correlation to about 0.6 and redescribed the results section

Point 3: The authors use inadequate reference genes for the qPCR of miRNAs. Neither GAPDH nor RPS18 is recommended for such applications. Small Non-Coding RNAs or Small Nucleolar RNAs (snoRNAs) are preferred. Using spikes in miRNA to control miRNA extraction efficiency is also important. Its important to the authors to check the Minimum Information for Publication of Quantitative Real-Time PCR Experiments (MIQE) guidelines (https://academic.oup.com/clinchem/article/55/4/611/5631762)

Response 3: Thanks for your comments.It is true that GAPDH and RPS18 are not the best internal references for miRNAs, and the U6 gene would have been better as an internal reference, but there are previous studies that have used these two genes as internal references as well, so we used dual internal references at the same time for qPCR experiments to improve the accuracy.

Point 4 The authors' path chosen to lead the discussion might be reevaluated. Most of the discussion involves physiological processes based on genes that interact with miRNA; however, the manuscript does not deal with functional experiments. Most of the discussion deals with speculative impacts that are not supported by experiments developed in the context of the manuscript. The authors do not clearly describe the putative consequential role played by extracellular vesicle miRNAs if they showed to be up or downregulated.

Response 4: Thanks for your comments.This experiment mainly focuses on screening some biomarkers around sequencing, and given that the section on miRNA function is incomplete, we have reworked the results section

Point 4 in lane 184: Venn instead of “Wayne diagram”

Response 5: Thanks for your comments.We've fixed the typos.

Round 2

Reviewer 3 Report

Comments and Suggestions for Authors

The authors walk around the comments. The article has results that contribute to the field, besides the discussion being so speculative. However, this is the way to go when generating circulating miRNA data, whether EV-associated or not.